# Predictive Embedding as Latent Action: Towards VLA Pretraining in the Wild

## Abstract

Vision-Language-Action models (VLAs) show promise for scalable robot learning, yet their progress is limited by small, narrow robot datasets. Human manipulation videos could provide richer learning material of skills, but current methods face a dilemma: either use expensive, precise labeled data with a limited scope, or abundant in-the-wild videos without hand tracking labels. We propose **PELA**, a pretraining framework that learns human motions by creating **P**redictive **E**mbeddings that align with **L**atent **A**ctions. Instead of trying to reconstruct every dynamics detail, PELA focuses on motion patterns that can be predicted from context and reflect real physical interaction. This creates a latent action space that captures motion dynamics across heterogeneous data sources. We build **UniHand-Mix**, a large hybrid dataset combining 5M carefully labeled lab recordings pairs with 2.5M pairs from in-the-wild human videos (7.5M total samples, >2,000 hours). This provides both reliable training signals and diverse real-world scenarios for large-scale learning. Our experiments show PELA generates realistic hand motions in both controlled and in-the-wild scenarios, and significantly improves downstream robot manipulation performance. The results demonstrate that predictive embeddings offer a practical route to scaling VLA pretraining using abundant human data.

## 1 Introduction

Robotics researchers have long sought foundation models enabling robots to perform diverse tasks. Recent Vision-Language-Action (VLA) models (Zitkovich et al., 2023; Black et al., 2024; Bjorck et al., 2025b) show promise by adapting Large Multimodal Models (LMMs) with robotics datasets (O'Neill et al., 2024; Khazatsky et al., 2024). However, robotic data remains orders of magnitude smaller and less diverse than datasets enabling breakthroughs in vision and language (Touvron et al., 2023; Li et al., 2024a). This scarcity, combined with cross-embodiment heterogeneity, creates a major obstacle for scaling VLA pretraining (Kim et al., 2025; Octo Model Team et al., 2024).

To address this data problem, recent works (Luo et al., 2025; Yang et al., 2025) have turned to cost-effective human demonstration videos, which contain diverse skilled hand movements valuable for VLA learning. By treating hand motions as actions, VLAs can learn skills unavailable in robot-only datasets. However, using human data creates a quality-variety trade-off: Lab-collected datasets (Fan et al., 2023; Chao et al., 2021; Hoque et al., 2025; Liu et al., 2022) provide precise 3D hand tracking but are limited to controlled tabletop scenarios, while in-the-wild human videos (Grauman et al., 2022; Perrett et al., 2025; Damen et al., 2022) offer tremendous variety and natural behaviors but lack precise action labels. This creates our core question: *how can we combine these complementary human data sources to scale VLA pretraining effectively?*

The challenge is that in-the-wild videos lack explicit action labels despite being rich in motion **dynamics** (how vision changes during interaction). Previous approaches capture this information indirectly through auxiliary visual representation learning (Nair et al., 2023; Radosavovic et al., 2023) or world models (Wu et al., 2024; Cheang et al., 2024), but their connection to action control remains weak. To extract more direct action information, recent works (Ye et al., 2025; Team, 2024b; Chen et al., 2024a) exploit inverse dynamics — the mapping from visual transitions to underlying actions — producing **latent actions** that form a **latent action space**. As shown in Fig. 1 (left), these approaches couple an inverse dynamics model (IDM) with a forward dynamics model (FDM): the IDM infers latent actions from current and future frames, while the FDM reconstructs future

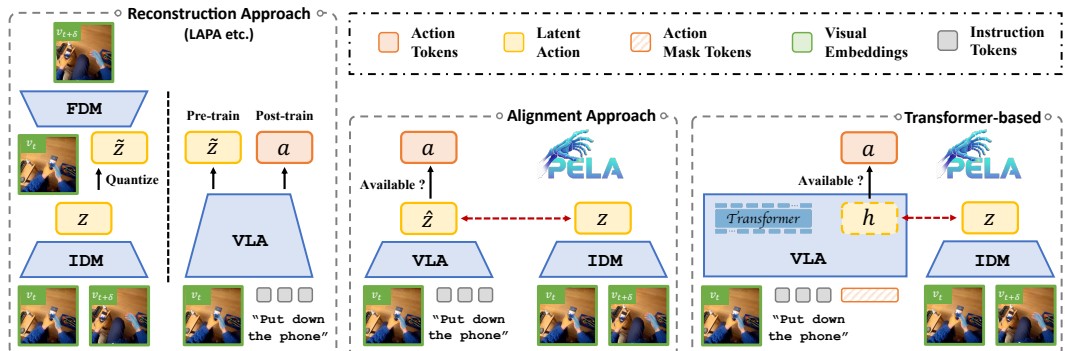

Figure 1: **Comparison of VLA's latent action paradigms with human videos. (left)** Prior reconstruction-based methods like LAPA (Ye et al., 2025) rely on multi-stage pipelines extracting latent actions via dynamics reconstruction as pseudo-labels. **(middle)** Our PELA introduces predictive embeddings aligned with latent actions. **(right)** Transformer-based PELA implementation where intermediate hidden states serve as the predictive embeddings to align with latent actions, while output tokens use available action labels as supervision.

frames from latent actions and current frames, constraining latent actions to encode action-relevant information. However, for fine-grained human manipulation, this paradigm is problematic because the mapping from latent to real actions relies heavily on the FDM's ability to model future frames, but human hand motions are so subtle and variable that this modeling is unreliable, often introducing noise rather than stable grounding.

In this paper, we rethink human actions for VLA pretraining. Inspired by how humans learn manipulation through key transferable patterns rather than memorizing details, we propose PELA (**P**redictive **E**mbedding as **L**atent **A**ction), which captures these patterns through constrained embeddings in a latent action space. Unlike methods requiring full reconstruction (Fig.1, left), PELA (Fig.1, middle) introduces lighter supervision where the VLA derives an embedding $\hat{z}$ satisfying two objectives: (1) soft alignment with IDM-derived latent actions $z$, and (2) robust mapping to ground-truth action $a$. This "predictive embedding" carries real action information while connecting to inverse dynamics. When aligned with predictive embeddings, latent actions capture essential action cues through real action supervision while filtering unnecessary visual details by avoiding full FDM reconstruction This alignment works consistently regardless of explicit action label availability, constructing a unified embedding space for learning from both lab-annotated and in-the-wild human videos.

To realize this design, we instantiate PELA on a Transformer-based VLA. As shown in Fig. 1 (right), the VLA predicts action tokens from action mask tokens while aligning intermediate hidden states $h$ with latent actions from IDM — these hidden states serve as predictive embeddings connecting real actions and latent actions. When we adapt the pretrained VLA for robot tasks, we feed these predictive embeddings into a flow-matching head (Lipman et al., 2022), efficiently transferring the pretrained latent action space to robot tasks.

To enable large-scale pretraining, we construct **UniHand-Mix**, combining lab-recorded videos with precise hand tracking and in-the-wild videos without tracking. Following Luo et al. (2025), we process 1,000 hours of lab videos to create 5M instruction-tuning samples with explicit motion labels, then extend to unconstrained settings by filtering manipulation clips from 1123 hours of Ego4D videos (Grauman et al., 2022), adding 2.5M in-the-wild samples. Built on InternVL3-2B (Chen et al., 2024b), PELA is evaluated on realistic hand motion generation and robot manipulation tasks. Results show PELA significantly improves hand motion generation in wild scenarios while maintaining lab performance, outperforming similar-size VLA models on Libero (Liu et al., 2023) and RoboCasa (Nasiriany et al., 2024) benchmarks, and remains competitive with larger models.

Our key contributions are: (1) **Novel latent action paradigm:** We introduce predictive embeddings, enabling VLAs to learn from both labeled and unlabeled human videos at unprecedented scale. (2) **Large-scale hybrid dataset:** We create UniHand-Mix, extending UniHand with 2.5M in-the-wild human manipulation samples, providing greater diversity for VLA pretraining. (3) **Strong empirical results:** We demonstrate a VLA that substantially improves generalization in hand motion generation with SoTA performance among similar-size models on robot tasks.

## 2 RELATED WORK

**Vision-Language-Action Model Pretraining.** There has been growing interest in building VLAs by adapting pretrained vision-language architectures for robotics. Using large-scale robot datasets (Padalkar & et al., 2023; Khazatsky et al., 2024; Bu et al., 2025), researchers have taken two main approaches: one convert robot actions into discrete tokens and trains like autoregressive LLMs (Brohan et al., 2023; Zitkovich et al., 2023; O'Neill et al., 2024; Kim et al., 2025; Pertsch et al., 2025; Zhong et al., 2025), while another uses diffusion- or flow-matching methods to generate continuous actions (Octo Model Team et al., 2024; Black et al., 2024; Intelligence et al., 2025; Bjorck et al., 2025b; Driess et al., 2025; Liu et al., 2024; Wen et al., 2025; Shukor et al., 2025). In our work, we use tokenization during pretraining to learn general action patterns, then switch to flow-matching for precise robot control during deployment. However, robot data alone is limited, so researchers have turned to human videos as a more scalable resource. Some works learn indirectly from human videos by extracting generic visual representations or building world-model (Bjorck et al., 2025b; Radosavovic et al., 2023; Nair et al., 2023; Wu et al., 2024; Cheang et al., 2024), or creating latent action spaces (Ye et al., 2025; Team, 2024a; Chen et al., 2024a). While promising, these approaches struggle to connect abstract representations to actual physical actions. Other approaches use lab-collected human data with precise hand tracking (Hoque et al., 2025; Banerjee et al., 2025; Liu et al., 2022; Chao et al., 2021) to explicitly model human hand movements (Luo et al., 2025; Yang et al., 2025; Singh et al., 2025a). This offers stronger physical grounding but requires expensive data collection and limits task diversity. Our work bridges these two approaches. We introduce a new method for constructing latent actions by aligning predictive embeddings with motion dynamics, allowing us to combine the scalability of learning from unlabeled videos with the precision of explicitly annotated trajectories. Our design reduces annotation cost while maintaining physical grounding, enabling VLA pretraining to scale more effectively across heterogeneous human datasets.

**Learning from Videos.** Our work also connects to research on *learning from videos* (LfV) (Yang et al., 2015; Seo et al., 2022), where studies extract behavioral knowledge from unlabeled videos using various techniques: masked auto-encoding which predicts missing parts of videos (Radosavovic et al., 2023; Xiao et al., 2022), temporal contrastive learning which learn to distinguish different video clips (Li et al., 2024b; Nair et al., 2023), video prediction which forecasts future frames (Seo et al., 2022; Luo et al., 2024), or inverse dynamics modeling which infers actions from visual changes (Baker et al., 2022; Schmeckpeper et al., 2021; Ye et al., 2022). For robot control specifically, some works exploit carefully matched human–robot pairs to enable imitation policies (Kareer et al., 2024; Singh et al., 2025b; Luo & Lu, 2025; Zheng et al., 2025b), but this approach only scales to specific task sets. When using unconstrained human videos with robot actions (Team, 2024a; Ye et al., 2025), effectiveness is limited due to unclear alignment between human and robot behaviors. Our approach differs by leveraging human hand tracking as a rich source of physical alignment, with the human hand serving as the most extensively annotated manipulator available. We combine these precise annotations with diverse in-the-wild videos within a unified pretraining framework, achieving both physical grounding and task diversity.

## 3 PRELIMINARIES

We introduce preliminaries for PELA, detailing human hand movement modeling during VLA pretraining and our hybrid data formulation with optional pose annotations.

Following Luo et al. (2025), we convert human manipulation data for VLA pretraining using MANO parameters (Romero et al., 2017) as our hand representation. Each dataset entry contains a video $v = \{v_1, v_2, \ldots, v_T\}$, textual instruction $x$, and hand pose sequence $\mathcal{M} = \{m_1, m_2, \ldots, m_T\}$. Each $m_t$ uses MANO parameters $(\theta_t, \mathbf{r}_t, \tau_t, \beta_t)$ for relative joint angles, global wrist rotation, wrist translation, and hand shape. We exclude $\beta_t$ in practice since it keeps static over time and does no help for dynamic understanding. For VLA pretraining, we convert continuous hand movements into discrete tokens by dividing pose sequence into fixed-length chunks and converting each into $K$ motion tokens via GRQ-With chunk length $\delta$ and $N = T/\delta$ chunks, the training sequence becomes: $[x, v_1, A_1, A_2, \ldots, A_N]$, where $v_1$ is the first video frame and $A_i$ denotes the $i$-th tokenized motion chunk. This format aligns with the VLA pretraining paradigm, allowing to generate hand motions based on visual and language inputs. Our VLA $f_\Theta$ learns to maximize motion token likelihood given

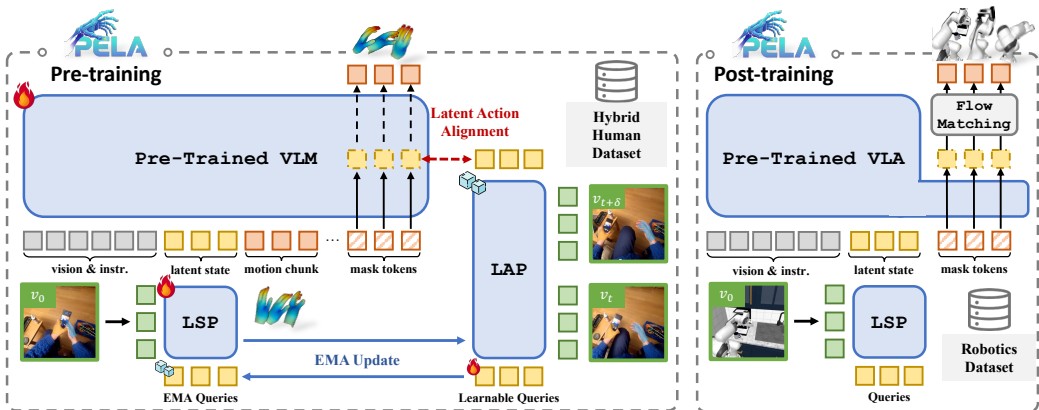

Figure 2: **The PELA framework**. **Pre-training (left)**: Hidden states of masked motion chunks serve as predictive embeddings to align with latent actions from boundary frames. The Latent Action Perceiver (LAP) maps boundary frames to latent action space, providing supervision without action labels. A parameter-shared Latent State Perceiver (LSP) injects initial frame context, with LAP and LSP linked via decoupled EMA update for stability. **Post-training (right)**: The predictive embeddings are fed into a flow-matching head for robot task transfer.

the previous context:

$$\max_{\Theta} \sum_{i=1}^{N} \log p(A_i \mid A_{<i}, v_1, x; \Theta). \tag{1}$$

Following Luo et al. (2025), we include auxiliary training tasks, generating text descriptions of motions (motion description generation) and continuing partial motion sequences (motion continuation). These tasks help the model learn richer token meanings before adapting to robot tasks while keeping the formulation of generating motions chunk by chunk.

Given limited availability of hand tracking data, we use both annotated and in-the-wild videos. Here, we define: (1) Annotated dataset $\mathcal{D}_A = \{(x^{(i)}, v^{(i)}, \mathcal{M}^{(i)})\}_{i=1}^{N_A}$ with instructions, videos, and hand poses. (2) Unannotated dataset $\mathcal{D}_U = \{(x^{(j)}, v^{(j)})\}_{j=1}^{N_U}$ with only instructions and videos. Our goal is jointly training on the hybrid dataset $\mathcal{D} = \mathcal{D}_A \cup \mathcal{D}_U$, leveraging both the precision of annotated data and the diversity of in-the-wild videos.

## 4 METHODOLOGY

As illustrated in Fig. 2, PELA builds on a Transformer-based vision-language model that processes visual tokens, instruction tokens, and motion tokens together. The key innovation is using intermediate hidden states as *predictive embeddings*, which act as a bridge between explicit hand motion labels and latent actions derived from inverse dynamics information. Through this bridging role, predictive embeddings capture both motion information and visual dynamics, deriving a unified latent action space that works consistently across both lab-collected and in-the-wild human data.

### 4.1 ALIGNING PREDICTIVE EMBEDDINGS WITH LATENT ACTIONS

For each motion token $a_{i,k}$ in a tokenized chunk $A_i$, we extract its hidden state from a preselected attention layer, denoted as $h_{i,k} \in \mathbb{R}^d$. These hidden states become our *predictive embeddings* and are shaped by two complementary signals: (1) supervision from motion labels, and (2) alignment with latent actions. Below we detail how to exploit these two signals, respectively.

**Masked Chunk Prediction (MCP).** To capture motion information based on hand tracking labels, we introduce MCP, an objective of chunk-level masked token modeling similar to action prediction in GR-1 (Wu et al., 2024). During pretraining, we replace all motion tokens in a chunk with [MASK] placeholder and use bidirectional attention within the chunk. This allows the model to understand

relationships between motions within a chunk. The model learns by predicting the original tokens:

$$\mathcal{L}_{\text{MCP}} = -\sum_{i=1}^{N}\sum_{k=1}^{K} \log p_{\Theta}\big(a_{i,k} \mid A_{<i}, v, x\big). \tag{2}$$

Although we sum over individual tokens, the bidirectional attention ensures that all motions $a_{i,k}$ in a chunk $A_i$ are modeled jointly, creating structured representations of chunk-level movement patterns.

**Latent Action Perceiver (LAP).** To align predictive embeddings with visual dynamics, we introduce LAP, similar to Jaegle et al. (2021); Shridhar et al. (2023), to act as an inverse dynamics model. For each chunk, LAP takes the start and end frames $(v_t, v_{t+\delta})$ and produces $K$ latent action vectors $\{z_{i,1}, \ldots, z_{i,K}\}$ using a fixed set of learnable query vectors. These latent actions capture the dynamics of short-horizon transition without needing to reconstruct full video frames. We then align our predictive embeddings $h_{i,k}$ with these latent actions $z_{i,k}$:

$$\mathcal{L}_{\text{Align}} = \sum_{i=1}^{N}\sum_{k=1}^{K} \|h_{i,k} - z_{i,k}\|_1. \tag{3}$$

To learn effective predictive embeddings, neither component works well alone. LAP by itself cannot guarantee transferable representations, and motion labels alone cannot capture visual dynamics. Predictive embeddings bridge them together: motion patterns from MCP and visual dynamics from latent action alignment are jointly compressed into the predictive embeddings. This dual constraint anchors predictive embeddings as a unified latent action space that works consistently across hybrid data types.

## 4.2 Joint Perceiver Implementation with Decoupled Updates

To capture short-horizon dynamics more effectively, we use features from a pretrained visual encoder (e.g., DINOv3 (Siméoni et al., 2025) or V-JEPA2 (Bardes et al., 2024)) as inputs to our latent action module. However, since these visual features originate from a model different from the VLM backbone, they may reside in a misaligned representation space. Simply connecting them to predictive embeddings could lose important information or create latent actions that don't match the model's context. To solve this, we introduce an additional **Latent State Perceiver (LSP)** to pair with the **Latent Action Perceiver (LAP)** (Fig. 2). While LAP processes boundary-frame features to generate latent actions, LSP connects the VLM's predictive context to the same latent action space to alleviate the predictive burden.

A central challenge is that the alignment loss directly ties two heterogeneous modules, risking collapse if one dominates the optimization. To address this, we decouple the updates of the latent Perceiver's backbone and its learnable queries. The backbone is optimized with the gradient from the LSP, ensuring that visual features are consistently mapped into the predictive embedding space shaped by MCP and latent-action alignment. In contrast, the queries are optimized with the gradient from the LAP, allowing them to specialize in anchoring latent actions with explicit action cues. To maintain consistency without overfitting to either side, we adopt an asymmetric EMA update: backbone weights are propagated from LSP to LAP, while query weights are propagated from LAP to LSP. This design ensures that latent actions remain both predictable from context and anchored with action cues, enabling stable integration of pretrained visual features into PELA's unified latent action space.

## 4.3 Training Details

**Pretraining on Hybrid Data.** Once we establish predictive embeddings as a bridge to latent actions, we can use the same approach for videos without hand tracking labels. For unlabeled videos, we skip MCP since no labels exist and only align predictive embeddings with visual dynamics from LAP. This allows in-the-wild human manipulation videos to contribute useful learning signals despite lacking precise annotations. As a result, we combine both labeled and unlabeled data using a hybrid training objective.

$$\mathcal{L} = \mathbf{1}_{\text{labeled}} \cdot \mathcal{L}_{\text{MCP}} + \lambda \mathcal{L}_{\text{Align}}, \tag{4}$$

where $\mathbf{1}_{\text{labeled}}$ is an indicator that only activates MCP when hand tracking labels are available. This means predictive embeddings learn from both sources: motion labels (when available) and dynamics (always), allowing PELA to scale pretraining across heterogeneous data.

**Post-training with Flow Matching.** After pretraining on hybrid human data, PELA establishes a unified latent action space through aligned predictive embeddings. For robot manipulation tasks, we transfer this space to robot-specific actions a **flow-matching head** that adapts predictive embeddings to the robot action space (Fig. 2, right). Given predictive embeddings $\{h_{i,k}\}$ from the pretrained VLA backbone, we feed them as a conditional input into the policy head, which is based on a Diffusion Transformer (DiT) with alternating self-attention and cross-attention modules. Self-attention processes the robot's proprioceptive state and noised actions, while cross-attention fuses this with predictive embeddings $\{h_{i,k}\}$, injecting the general dynamics knowledge learned from pre-training into action generation. During post-training we employ a flow-matching objective. For a ground-truth action chunk $A_t$, we construct a noised action $A_t^\tau = \tau A_t + (1 - \tau)\epsilon$ based on a timestep $\tau \in$ and standard Gaussian noise $\epsilon$. The model $V_\theta$ learns to predict the denoising vector field $\epsilon - A_t$:

$$\mathcal{L}_{\text{FM}} = \mathbb{E}_{\tau, \epsilon, A_t} \left[ \|V_\theta(\{h_{i,k}\}, A_t^\tau, q_t) - (\epsilon - A_t)\|_2^2 \right], \tag{5}$$

where $q_t$ is the robot's proprioceptive state. During inference, by applying the learned model $V_\theta$, we iteratively denoise a random initial action chunk using forward Euler integration with a fixed step number. This ultimately yields precise robot action commands, enabling PELA to efficiently transfer dynamics-rich knowledge from human data to downstream precise robot control.

## 5 UNIHAND-MIX: HYBRID HUMAN MANIPULATION DATASET

To scale PELA pretraining, we construct **UniHand-Mix**, a hybrid dataset unifying lab-annotated human manipulation data and in-the-wild videos.

**Lab-annotated Subset.** Similar to the pipeline proposed in Luo et al. (2025), each sample includes instruction, video, and MANO-based hand motion sequence. We provide three instruction-tuning types: *motion generation*, *motion description*, and *motion continuation*. Finally, we process approximately 1,000 hours of lab videos, yielding 5M+ instruction–video-motion samples.

**In-the-wild Subset.** We curate an in-the-wild subset from Ego4D (Grauman et al., 2022) to add diverse human interactions. Compared to controlled labs, egocentric recordings pose unique challenges: intermittent hand visibility from occlusions and irrelevant content like idle hands. To distill clean manipulation episodes, we design a two-stage pipeline. **(1) Visual Filtering**: We detect hand regions at the frame level using Potamias et al. (2025), and apply pose estimation using HaWoR (Zhang et al., 2025b), retaining high-confident clips with approximate hand supervision as a bridge between lab and in-the-wild data. **(2) Instruction Validation:** We use Gemini-2.5-flash to identify hand-centric activities, discarding clips with absent/idle/distractor hands while pairing remaining clips with auto-generated instructions. This yields 2.5M instruction–video pairs, enriching UniHand-Mix with broader task diversity and contextual priors for enhancing pretraining scalability.

We further present more details and some statistics in the Appendix B.

## 6 EXPERIMENTS

Our experimental study investigates three core questions: **(1) In-the-wild Learning:** Can PELA learn from in-the-wild human videos and demonstrate scaling? **(2) Downstream Transfer.** Does predictive embedding improve robot task performance? **(3) Ablation Validation.** Are the proposed alignment designs effective?

### 6.1 IMPLEMENTATION DETAILS

**Pre-training.** PELA uses `InternVL3-2B` backbone (Chen et al., 2024b) with DINOv3 (Siméoni et al., 2025) or V-JEPA2 (Bardes et al., 2024) as the visual encoder feeding into the latent perceiver. Each 15-frame motion chunk is tokenized by GRQ-VAE into 128 discrete tokens (codebook size 8192). In-the-wild data is temporally slowed by a factor of $0.5$, to account for the differing action speeds with lab-collected data. The 19th attention layer (out of 28) outputs predictive embeddings for alignment with inverse-dynamics signals. The unified loss is optimized with $\lambda = 0.5$, AdamW, cosine learning rate scheduling and mixed precision. More details are in Appendix A.

Table 1: Comparison of hand motion generation and prediction tasks on both Lab and Wild splits.

| Model | MPJPE (cm) ↓ | | PA-MPJPE (cm) ↓ | | MWTE (cm) ↓ | | MDE (cm) ↓ | |
|---|---|---|---|---|---|---|---|---|
| | Lab | Wild | Lab | Wild | Lab | Wild | Lab | Wild |
| **# Next Token Prediction** | | | | | | | | |
| Being-H0-2B | 7.61 | 16.91 | 1.34 | 3.81 | 6.03 | 14.54 | 7.16 | 18.33 |
| Being-H0-2B+dino | 7.54 | 15.14 | 0.90 | 2.78 | 5.85 | 13.65 | 6.95 | 17.17 |
| **# Masked Chunk Prediction** | | | | | | | | |
| PELA w/o align | 7.72 | 15.73 | **0.89** | 2.34 | 6.18 | 14.02 | 8.09 | 16.23 |
| PELA-dino | 7.16 | **11.02** | 0.91 | **1.12** | **5.77** | **9.79** | 7.24 | **11.04** |
| PELA-vjepa | **7.05** | 11.54 | 0.94 | 1.32 | 5.85 | 10.04 | **6.73** | 11.87 |

**Post-training.** After pre-training, we fine-tune PELA on LIBERO and RoboCase using a flow-matching head. Training uses batch size 128, learning rate 1e-4 with cosine decay and 5% warm-up. Only language model parameters are unfrozen and the vision encoder remains frozen. We train a GR00T-N1.5 baseline for comparison with identical settings. Success rate is the primary metric.

**Simulation Benchmark.** In this work, we use the following two benchmark for robotic task evaluation: **(1) LIBERO:** We independently fine-tune our model on four task suites (Spatial, Object, Goal, and Long). For each suite, we use $< 50$ expert demonstrations and train our model for 30,000 steps. **(2) RoboCasa:** To assess the model's performance on different data sources, we conduct two experiments on the atomic tasks: the first one trained with 50 **human demonstrations** per task, another with 50 **synthetic demonstrations** per task. Both train for 60,000 steps.

## 6.2 HAND MOTION GENERATION

We assess our model on the hand motion generation task which predicts a motion chunk given an initial visual input and instruction. Performance on this task directly reflects how well a model captures human strategy priors from the pretraining data. We report results on two testing splits of UniHand-Mix: **Lab split** held-out from lab-annotated data, which measures fidelity under precise supervision; and **Wild split** curated from Ego4D with HaWoR annotations, which measures the model's generalization to unconstrained human interactions.

**Model Variants and Baselines.** We report two variants of PELA: **PELA-dino** and **PELA-vjepa** (using DINOv3 or V-JEPA2 visual feature). We also use three baselines for comparison: **Being-H0-2B** (Being-H0 reproduction on our annotated subset with a InternVL-2B backbone), **Being-H0-2B+dino** (add DINOv3 features like PELA's LSP), **PELA w/o align** (PELA-dino without LAP to isolate the alignment effect).

**Evaluation metrics.** We report four metrics: **(1) MPJPE**, the mean Euclidean distance between predicted and ground-truth 3D joints for spatial accuracy measurement; **(2) PA-MPJPE**, the MPJPE after rigid alignment for relative pose fidelity measurement; **(3) MWTE**, the average offset of wrist trajectories for global trajectory fidelity measurement; **(4) MDE**, the error of final displacement direction from the initial wrist position for motion trend consistency measurement. These metrics capture both local pose accuracy and global trajectory realism.

**Main Results.** Tab. 1 shows that both PELA variants outperform baselines on most metrics. Gains are modest on the Lab split but substantial promotion is shown on the Wild split, demonstrating PELA's scaling to diverse scenarios. While all models degrade on the Wild split compared to the Lab split due to higher variability and complexity, PELA achieves much smaller gaps, confirming the benefit of large-scale in-the-wild videos.

**Design Insights.** Through a set of controlled comparisons, we can further dissect the reasons for the improvements. Being-H0-2B+dino vs. Being-H0-2B shows that adding self-supervised visual features yields moderate gains, while its similar performance to PELA w/o align indicates that prediction paradigm (next-token vs. masked chunk) isn't decisive. The gap between PELA w/o align and full PELA variants confirms the alignment mechanism as the key contributor for effective in-the-wild video use. The comparable performance of PELA-dino and PELA-vjepa shows PELA robustly constructs a unified latent action space from self-supervised visual features.

Table 2: Results on LIBERO benchmark showing success rates (%) by task category and overall average. Models are grouped by size: >3B parameters (top) and ≤3B parameters (bottom). The results of LAPA⋆ are reported in UniVLA. UniVLA-human[†] uses only human pretraining data. UniVLA-full[††] incorporates Bridge-V2 (Walke et al., 2023) for pretraining.

| Model | Spatial | Object | Goal | Long | Average |
|---|---|---|---|---|---|
| **# > 3B Backbones** | | | | | |
| LAPA⋆ (Ye et al., 2025) | 73.8 | 74.6 | 58.8 | 55.4 | 65.7 |
| OpenVLA (Kim et al., 2025) | 84.7 | 88.4 | 79.2 | 53.7 | 76.5 |
| TriVLA (Liu et al., 2025) | 91.2 | 93.8 | 89.8 | 73.2 | 87.0 |
| 4D-VLA (Zhang et al., 2025a) | 88.9 | 95.2 | 90.9 | 79.1 | 88.5 |
| UniVLA-human[†] (Team, 2024a) | 91.2 | 94.2 | 90.2 | 79.4 | 88.7 |
| UniVLA-full[††] (Team, 2024a) | **96.5** | **96.8** | **95.6** | **92.0** | **95.2** |
| **# ≤ 3B Backbones** | | | | | |
| Diffusion Policy (Chi et al., 2023) | 78.3 | 92.5 | 68.3 | 50.5 | 72.4 |
| Octo (Octo Model Team et al., 2024) | 78.9 | 85.7 | 84.6 | 51.1 | 75.1 |
| UniACT (Zheng et al., 2025a) | 77.0 | 87.0 | 77.0 | 70.0 | 77.8 |
| SpatialVLA (Qu et al., 2025) | 88.2 | 89.9 | 78.6 | 55.5 | 78.1 |
| DiT Policy (Hou et al., 2024) | 84.2 | 96.3 | 85.4 | 63.8 | 82.4 |
| ThinkAct (Huang et al., 2025) | 88.3 | 91.4 | 87.1 | 70.9 | 84.4 |
| Being-H0-2B (Luo et al., 2025) | 86.6 | 92.8 | 89.6 | 70.4 | 84.9 |
| GR00T N1.5 (Bjorck et al., 2025b) | 91.4 | 97.6 | 94.0 | 85.6 | 92.1 |
| PELA w/o align | 89.4 | 91.2 | 90.0 | 72.6 | 85.8 |
| PELA-vjepa | **91.6** | **98.2** | 94.4 | 84.2 | 92.1 |
| PELA-dino | 90.4 | 96.4 | **95.2** | **87.2** | **92.3** |

Table 3: Results on the RoboCasa benchmark. We report success rates (%) across task categories and the overall average for all the models.

| Data Source | GR00T N1.5 | Being-H0-2B | PELA w/o align | PELA-vjepa | PELA-dino |
|---|---|---|---|---|---|
| Synthetic Data | 20.83 | 23.83 | 24.50 | 27.17 | 27.58 |
| Human Demonstrations | 35.17 | 31.33 | 31.75 | 34.92 | 35.42 |
| Average | 28.00 | 27.58 | 28.13 | 31.04 | 31.50 |

## 6.3 SIMULATION RESULTS

**LIBERO.** We fine-tune PELA on the four task suites from the LIBERO benchmark and compare its performance against state-of-the-art methods, particularly those with backbones of a similar scale (<3B parameters). As shown in Table 2, our PELA models, utilizing either V-JEPA or DINO visual features, demonstrate top-tier performance. Specifically, PELA-dino establishes a new state-of-the-art average success rate of 92.3% among models in this category, outperforming strong baselines like GR00T N1.5 (92.1%). This strong performance is particularly notable in the most challenging Long task suite, where PELA-dino achieves the highest score of 87.2%, highlighting our model's superior generalization capabilities on long-horizon tasks, which are notoriously difficult. Furthermore, the importance of our proposed alignment strategy is validated by the PELA w/o align ablation, which sees a significant performance drop to an average of 85.8%. This result underscores that our alignment mechanism is critical to the model's success. Notably, our lightweight PELA model also outperforms some larger 7B-parameter models, showcasing the efficiency and effectiveness of our approach.

**RoboCasa.** As shown in Table 3, our PELA model comprehensively outperforms all baseline models, including GR00T N1.5. PELA-dino achieves an average success rate of 31.50%, significantly higher than the 28.00% from GR00T N1.5, demonstrating the superiority of our pre-training methodology.

The advantage of PELA is particularly pronounced on synthetic data. PELA-dino's success rate (27.58%) far exceeds that of GR00T N1.5 (20.83%), showing an improvement of nearly 7 percentage points. This strongly suggests that our model's latent action space and pre-training priors can more effectively distill useful control strategies from large-scale, albeit potentially biased, synthetic data. On human demonstrations, our model also delivers outstanding performance, with both PELA-dino (35.42%) and PELA-vjepa (34.92%) achieving leading results that surpass GR00T N1.5 (35.17%) and all other compared methods. This demonstrates that PELA not only excels in overall performance but

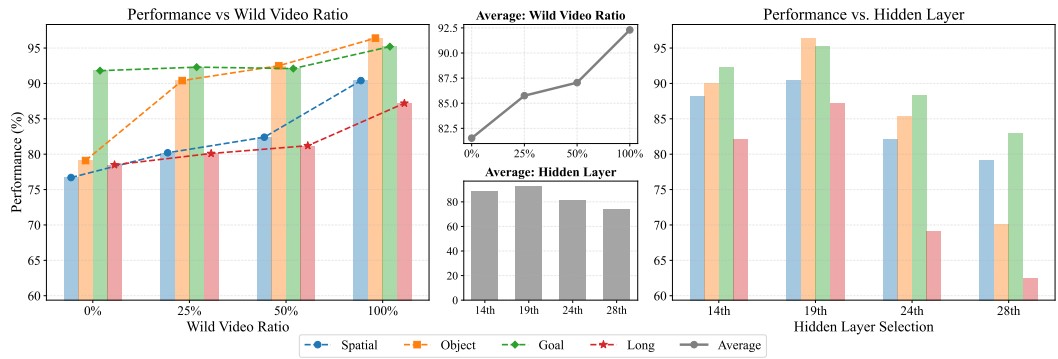

Figure 3: Ablation studies on PELA-dino evaluated on LIBERO. **Left:** performance across various the proportion of in-the-wild data used during pretraining (0%, 25%, 50%, 100%). **Right:** performance when feeding hidden states from different backbone layers (14, 19, 24, 28) into the flow-matching head during adaptation. **Middle:** average success rate across task suites for these variants.

also possesses strong data adaptability, learning efficiently to achieve the best performance regardless of whether the data source is real human operations or synthetic algorithms. Furthermore, the ablation result of PELA w/o align (28.13%) again confirms the criticality of our alignment strategy, as its removal causes performance to regress to a level on par with the baseline.

### 6.4 ABLATION STUDIES

We conduct two ablations on the LIBERO benchmark with PELA-dino to validate (i) its scalability to unconstrained human videos, and (ii) the role of predictive embeddings compared with other generic hidden states.

**Scalability to in-the-wild data.** To assess how large uncurated videos contribute to PELA, we vary the proportion of in-the-wild data during pretraining while keeping lab-annotated data fixed (0%, 25%, 50%, and 100%). Results on LIBERO (Fig. 3, left) show a consistent improvement in downstream success rates as more in-the-wild data are added, confirming the strong scaling potential of our framework.

**Flow-matching input ablation.** We further test the effectiveness of different hidden states from the backbone when fed into the flow-matching head during downstream adaptation. Note that we only vary the layer providing inputs to flow-matching on the PELA-dino pre-trained with the 19[th] layer used for alignment. We compare hidden states from layers 14, 19, 24, and 28. As shown in Fig. 3 (right), using the 19[th] layer yields the best transfer performance, the 14[th] layer is slightly worse, while later layers degrade sharply. This might suggest that alignment concentrates generalizable cues in the selected layer, whereas deeper layers overfit to dataset-specific details, reducing their utility for downstream robot control.

## 7 CONCLUSION

In this work, we presented **PELA**, a pretraining framework that rethinks how to represent latent actions for scalable VLA pretraining with human videos. By introducing predictive embeddings aligned with latent actions, PELA moves beyond reconstruction-heavy paradigms and instead captures motion cues without detailed distortion. This design enables a unified latent action space that works consistently across lab-annotated and in-the-wild human data. To support pretraining at scale, we constructed **UniHand-Mix**, a 7.5M-sample dataset combining over 2,000 hours of annotated and unconstrained human videos. This hybrid resource provides both reliable physical anchors and broad task diversity, and our experiments demonstrate that PELA can effectively leverage both sources. Results show strong improvements on hand motion generation and gains on downstream robot manipulation benchmarks, surpassing existing VLA methods with comparable or larger model sizes. Overall, our study highlights predictive embeddings as a practical and scalable route for bridging human video data with robotic learning. We believe this perspective opens opportunities for learning from even larger and more diverse human corpora.

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

# Appendix

## A ADDITIONAL IMPLEMENTATION DETAILS

### A.1 MOTION TOKENIZATION

Our PELA architecture follows the Being-H0 design and is built on InternVL3-2B as the vision-language backbone with 28 attention layers. To incorporate the motion modality, we tokenize *15-frame motion chunks*. Each chunk is decomposed into *wrist* and *finger* motions, which are separately quantized into 64 tokens each, yielding 128 tokens per chunk. The codebook size of each part is 4096, learned via a *GRQ-VAE* self-supervised clustering algorithm to capture general motion patterns. The motion tokens are then wrapped with two **special tokens** <mot> and </mot>, forming the following unified format:

<mot> {wrist_motion_tokens} {finger_motion_tokens} </mot>

These special tokens serve as explicit delimiters, helping the VLM distinguish motion chunks from other modalities such as vision or instruction tokens. For sequences containing **both hands**, we interleave the left-hand and right-hand chunks along the temporal axis to preserve synchrony while maintaining modality distinction.

### A.2 LATENT ACTION PERCEIVER (LAP) & LATENT STATE PERCEIVER (LSP)

**Structure.** The Latent Action Perceiver (LAP) and Latent State Perceiver (LSP) share the same structure and weights. We adopt a 2-layer Perceiver module, where each layer consists of:

1. A **cross-attention block**, using visual features as key-value pairs and learnable latent queries for information extraction.

2. A **self-attention block** applied on the extracted latent tokens for within-latent aggregation.

The resulting features are passed through a 2-layer MLP to map them into the VLM embedding space. Since videos may contain two hands, latent actions originating from the same Perceiver must align to different motion chunks. To maximize sharing while allowing differentiation, we double the channel dimension of the Perceiver's MLP and split it into two hand-specific heads. Each sample dynamically selects the left- or right-hand head depending on the active motion stream. Specially, for consistency between the Latent Action Perceiver (LAP) and the Latent State Perceiver (LSP), both modules are designed to process pairs of frames. While LAP takes the boundary frames $(v_t, v_{t+\delta})$ of each motion chunk, in LSP we duplicate the initial frame $(v_0, v_0)$. This ensures that differences between LAP and LSP stem solely from their input semantics (dynamics vs. context), rather than from architectural mismatch.

**EMA update for Perceiver.** A key challenge in aligning predictive embeddings with latent actions is that the Latent Action Perceiver (LAP) and Latent State Perceiver (LSP) process heterogeneous signals (visual boundary frames vs. predictive context). Direct joint optimization often leads to instability or collapse. To stabilize training, we adopt a decoupled exponential moving average (EMA) update between the two modules. Concretely, let $\theta_b^{\text{LAP}}, \theta_q^{\text{LAP}}$ denote the backbone and query parameters of LAP, and $\theta_b^{\text{LSP}}, \theta_q^{\text{LSP}}$ those of LSP. We update them as:

$$\theta_b^{\text{LAP}} \leftarrow \alpha \, \theta_b^{\text{LAP}} + (1 - \alpha) \, \theta_b^{\text{LSP}}, \tag{6}$$

$$\theta_q^{\text{LSP}} \leftarrow \alpha \, \theta_q^{\text{LSP}} + (1 - \alpha) \, \theta_q^{\text{LAP}}, \tag{7}$$

where $\alpha \in [0, 1)$ is the EMA coefficient. This asymmetric design ensures that the LAP backbone stays consistent with the predictive context shaped by MCP and alignment losses, while the LSP queries gradually inherit the action-grounding capability from LAP. In practice, we set $\alpha = 0.999$ to balance stability and adaptability.

### A.3 MASKED CHUNK PREDICTION (MCP)

During pretraining, we apply masked decoding to enforce predictive consistency. A naive masking strategy introduces a mismatch between training and inference, since all tokens in a chunk are replaced by [MASK] during training, but in inference, motion chunks are generated sequentially.

To mitigate this gap, we design a **hybrid masking scheme**:

- For each sequence with $N$ chunks, we randomly select one chunk as the main prediction target.
- Chunks before the target are kept intact (no masking).
- Inside the target chunk, each token is masked with a random ratio uniformly sampled from $\{0.05, 0.15, \ldots, 1.0\}$.
- Tokens in chunks after the target are masked with a fixed 5% probability to provide additional supervision without distorting context.

This ensures that the main prediction chunk has aligned context during both training and inference.

**Unlabeled videos (no motion tokens).** For in-the-wild *video-only* samples that lack motion tokens, the entire motion chunk $A_i$ is replaced by [MASK] placeholders. In this case, the MCP term is inactive, and training proceeds solely via alignment to latent actions from LAP (i.e., only $\mathcal{L}_{\text{Align}}$ is applied). This keeps the interface unified while still learning predictive embeddings that are aligned to dynamics without requiring explicit motion labels.

**Inference.** For motion generation, we decode the current chunk **multiple times**, each time decode $\sim 5\%$ of the tokens in the chunk; the outputs are ensembled to reduce approximation error. This retains the efficiency advantage over causal decoding, while downstream transfer still uses a **single forward pass** to extract predictive embeddings.

**Flow-matching head for robot adaptation.** For post-training adaptation, we directly follow the flow-matching head design in GR00T N1.5 (Bjorck et al., 2025a). Given predictive embeddings $\{h_{i,k}\}$ from the pretrained VLA backbone, we use them as conditional input to a Diffusion Transformer (DiT) policy head composed of alternating self-attention and cross-attention layers. We employ a DiT with 16 layers of 32-head attention blocks, and the hidden state dimension is 2048. Self-attention operates over the robot's proprioceptive states and noisy actions, while cross-attention integrates predictive embeddings $\{h_{i,k}\}$ to inject human-derived dynamics knowledge. The training objective is a flow-matching loss:

$$\mathcal{L}_{\text{FM}} = \mathbb{E}_{\tau,\epsilon,A_t} \left[ \|V_\theta(\{h_{i,k}\}, A_t^\tau, q_t) - (\epsilon - A_t)\|_2^2 \right], \tag{8}$$

where $A_t^\tau = \tau A_t + (1 - \tau)\epsilon$ is a noised action chunk, $q_t$ is the proprioceptive state, and $\epsilon \sim \mathcal{N}(0, I)$. During inference, we initialize with Gaussian noise and iteratively denoise using forward Euler steps, typically with a fixed schedule of $N$ steps (we use $N = 4$ by default). This process generates precise action sequences consistent with the pretrained latent action space, enabling efficient transfer of dynamics-rich human priors to robotic control tasks.

### A.4 TRAINING DETAILS

We optimize PELA with AdamW using a base learning rate of $3 \times 10^{-5}$, weight decay of $0.05$, and $\beta = (0.9, 0.95)$. The learning rate is warmed up for the first 5% steps and then decayed with a cosine schedule, while gradient clipping (max norm 1.0) is applied throughout. Training uses an effective batch size of 128 sequences, obtained from a per-GPU batch size of 16 with gradient accumulation across 8 GPUs, where each sequence contains a 15-frame motion chunk plus paired instructions and boundary frames. The hybrid loss combines masked chunk prediction and latent-action alignment with $\lambda = 0.5$, and for in-the-wild videos, only the alignment loss is applied. The perceiver modules are updated with an EMA coefficient of $\alpha = 0.999$ to stabilize training.

Pretraining is performed on the full 7.5M UniHand-Mix dataset for a single epoch, which requires 68 hours on 8 NVIDIA A800 (80GB) GPUs. For downstream adaptation with the flow-matching head,

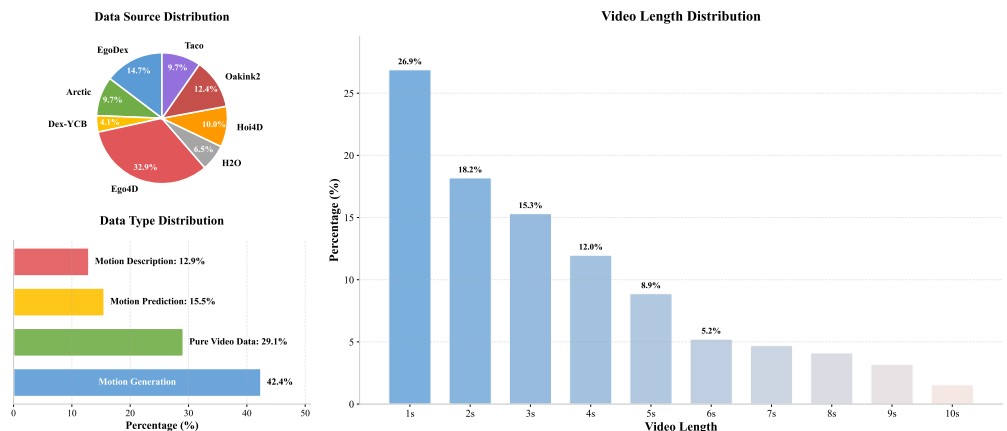

Figure 4: **Dataset statistics of UniHand-Mix.** (Left top) Source distribution across lab-collected and in-the-wild datasets. (Left bottom) Breakdown by task type. (Right) Distribution of clip lengths (1–10 seconds). Together, these statistics highlight the diversity and scale of UniHand-Mix, enabling scalable pretraining.

we freeze the visual modules and use a batch size of 128, learning rate of $1 \times 10^{-4}$ with cosine decay and 5% warm-up. On LIBERO tasks we run 30k steps (≈5 hours), while on RoboCasa tasks we run 60k steps (≈10 hours), using the same hardware configuration.

## B DATASET DETAILS

### B.1 DATA CURATION STEPS

**Lab-Collected Subset.** Following the UniHand Luo et al. (2025) pipeline, we curate a high-quality lab-collected subset with precise 3D hand annotations and dense task descriptions. (1) *Hand pose standardization:* All annotations are unified into the MANO parameter format (Romero et al., 2017). For datasets with mocap or SLAM labels, we directly convert to MANO. For datasets with only 3D joints, we fit MANO via gradient optimization. For raw RGB-only data, we apply HoWaR (Zhang et al., 2025b) for per-frame estimation, followed by temporal smoothing and left–right correction. (2) *Task labeling:* Videos are segmented into 10s chunks and annotated hierarchically. At the chunk level, we produce imperative instructions and concise summaries. At the second level, we annotate contact states, object properties, and hand–object interactions, including both two-handed and single-hand actions. (3) *Instructional data generation:* Based on these annotations, we construct multimodal tasks—motion generation, motion translation, and motion prediction—using 20 base templates per task type, diversified via Gemini-2.5-Pro. This establishes explicit grounding between vision, language, and motion.

**In-the-Wild Subset.** To complement lab data with naturalistic human behaviors, we curate an additional subset from Ego4D (Grauman et al., 2022). Unlike controlled setups, egocentric recordings bring challenges such as frequent hand occlusion and irrelevant non-manipulative segments. To extract meaningful manipulation episodes, we employ a two-stage filtering pipeline. First, *visual filtering:* hand regions are detected with an off-the-shelf detector (Potamias et al., 2025), and candidate clips are annotated with HaWoR (Zhang et al., 2025b) to obtain approximate pose supervision. Only high-confidence clips are retained, and the threshold is 0.65 for both. Second, *instruction validation:* Gemini-2.5-flash-lite is prompted to verify the presence of hand-centric activities, removing idle or distractor clips. Valid clips are then paired with automatically generated natural language instructions. This process yields roughly 2.5M instruction–video pairs, which, combined with 5M lab-annotated samples, form the UniHand-Mix dataset. The two subsets provide complementary strengths: lab data anchors physical accuracy, while in-the-wild data brings diversity and contextual richness, together enabling scalable pretraining.

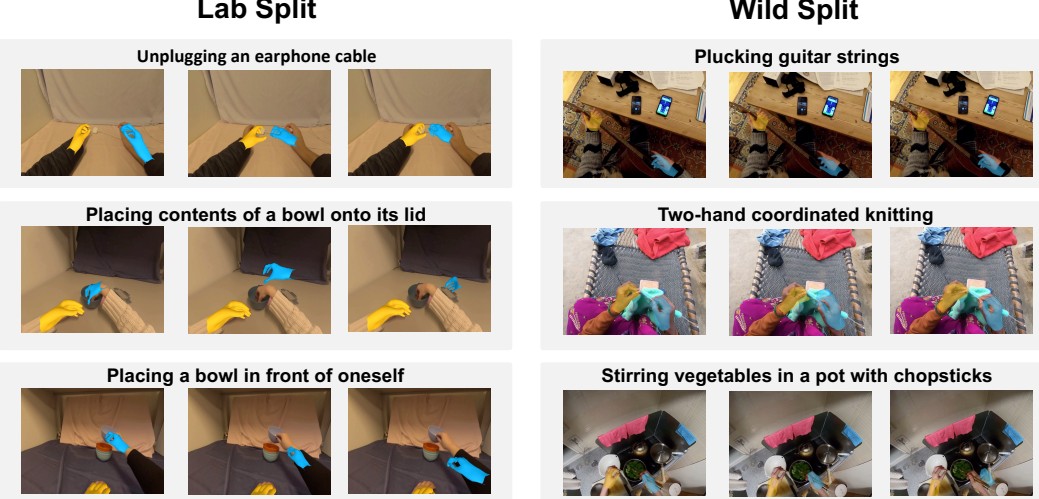

Figure 5: **Qualitative hand-motion generation on wild (left column) and lab (right column) scenes.** Left: (top) *Plucking guitar strings*; (middle) *Two-hand coordinated knitting*; (bottom) *Stirring vegetables in a pot with chopsticks*. Right: (top) *Unplugging an earphone cable*; (middle) *Placing contents of a bowl onto its lid*; (bottom) *Placing a bowl in front of oneself*. Colored overlays denote generated hand poses.

## B.2 DATASET STATICS

To provide a clearer picture of the curated UniHand-Mix dataset, we summarize its composition and distributional properties in Fig. 4. The dataset combines diverse sources, task types, and temporal scales:

- **Data Sources.** As shown in the pie chart (top left), UniHand-Mix integrates both lab-collected datasets (e.g., DexYCB, OakInk2, H2O, Arctic, EgoDex, TACO) and in-the-wild videos (Ego4D), yielding a balanced mix of precise motion annotations and naturalistic behaviors.

- **Data Types.** The bar chart (bottom left) shows the breakdown by task type: motion generation (42.4%), motion prediction (15.5%), motion description (12.9%), and pure video data without explicit motion tokens (29.1%).

- **Video Lengths.** The histogram (right) illustrates the distribution of clip lengths. While short clips (1–3 seconds) dominate, longer sequences (up to 10 seconds) are also included, ensuring coverage of both fine-grained and long-horizon manipulations.

## B.3 ADDITIONAL QUALITATIVE EXAMPLES

Figure 5 presents qualitative hand–motion generations covering both in-the-wild and lab–collected scenarios. On the **wild** side (left column), our model successfully handles diverse, unconstrained interactions such as *plucking guitar strings*, *two-hand coordinated knitting*, and *stirring vegetables in a pot with chopsticks*, demonstrating robust generalization to complex scenes and bimanual coordination. On the **lab** side (right column), the model produces precise and temporally consistent motions for *unplugging an earphone cable*, *placing contents of a bowl onto its lid*, and *placing a bowl in front of oneself*, reflecting accurate fine-grained control in structured settings. These results illustrate that PELA's predictive embeddings support both generalization in the wild and accuracy in controlled environments.

## C    USE OF LARGE LANGUAGE MODELS

The large language model (LLM) in this work is applied exclusively for text polishing. Its scope is limited to refining linguistic quality, coherence, and stylistic consistency, and it plays no role in the generation of data or the creation of substantive content.

