# OpenReview forum: "Predictive Embedding as Latent Action: Towards VLA Pretraining in the Wild"
_ICLR.cc/2026/Conference — ICLR 2026 Conference Withdrawn Submission_

### Official Review · Reviewer_E7mX · 2025-10-31

**Soundness:** 2
**Presentation:** 3
**Contribution:** 2
**Rating:** 4
**Confidence:** 4

**Summary:**

The paper proposes PELA (Predictive Embedding as Latent Action), a pretraining framework that:
- extracts predictive embeddings from mid-layer VLM features
- aligns them to latent actions predicted by an inverse-dynamics module using boundary frames
- reuses these embeddings in a flow-matching policy for robot control or hand motion prediction.

The data recipe mixes lab-annotated human-hand clips (with pose/motion labels) and in-the-wild videos (without labels). Two losses are used during pretraining: Masked Chunk Prediction (MCP) applied to lab clips, and an alignment loss applied to both lab and wild clips to construct a purportedly unified "action-centric" embedding space. The paper reports improved hand-motion generation and downstream control on LIBERO/RoboCasa, plus ablations on: (a) which embedding layer performs best for post-training, and (b) LIBERO performance vs proportion of in-the-wild data used during pretraining.

**Strengths:**

- **Relevant problem statement.** Aligning mid-layer predictive features to short-horizon latent actions (via boundary frames) tackles a limitation of pixel-reconstruction for fine hand motion: it provides an explicitly action-predictive signal to the backbone without requiring full future-frame synthesis.
- **Supervision from in-the-wild videos.** The alignment mechanism enables learning from unlabeled human manipulation footage and can be applied at scale. This is also supported by ablations on LIBERO performance.
- **Grounded latent-action learning.** When action labels are available, the method couples alignment with sequence prediction (MCP loss) so latent actions are anchored in actual motion trajectories. This differs from LAPA-style approaches that rely primarily on alignment to IDM targets without an explicit sequence-prediction objective on labeled data.

**Weaknesses:**

- **Ablation fairness.** I think "w/o align" mode likely drops all in-the-wild supervision. The pretraining has two losses (MCP on lab, alignment on lab+wild videos). The "w/o align" setting seems to eliminate the only loss used on wild clips, effectively turning it into a lab-only run. This conflates loss removal with data removal, making it impossible to attribute gains specifically to alignment rather than to having extra wild data.
- **No real-robot evidence.** All results are reported in simulation. For hand manipulation, practical deployment typically needs >15 Hz closed-loop control with low jitter. The paper does not report end-to-end inference latency, supported control frequency, or any on-robot trials. Without these, it is unclear whether the method is deployable for real robots/hands or only sim-viable.
- **No alignment-only run.** It remains unclear whether alignment alone is a sufficient supervisory signal. Without MCP, do embeddings still encode motion-useful structure directly from videos?
- **Alignment location is fixed during pretraining.** The paper ablates which embedding layer is best for post-training readout, but the alignment head is always attached to the same layer during pretraining. As a result, we can’t tell whether the gains come from where alignment shapes the backbone vs simply which layer is read out later. For instance, what would happen if alignment is done after the last transformer layer?
-  **Representation claims lack analysis.** As lab samples receive MCP+alignment while wild samples receive alignment only, the unified space claim needs some analysis that wild embeddings encode motion rather than merely being dragged toward lab. For instance, would it work for zero-shot motion decoding on wild videos?

**Questions:**

Please address the points raised in Weaknesses section.

---

### Official Review · Reviewer_Mjen · 2025-10-31

**Soundness:** 3
**Presentation:** 2
**Contribution:** 2
**Rating:** 2
**Confidence:** 4

**Summary:**

The paper introduces PELA, a new framework for pretraining VLA models using large-scale human manipulation videos. Instead of reconstructing full visual dynamics as in prior latent-action approaches, PELA learns predictive embeddings that align with latent actions derived from inverse dynamics. These embeddings capture motion patterns that are predictable and physically meaningful, enabling learning from both labeled lab data and unlabeled in-the-wild videos.

**Strengths:**

1. The paper redefines how Vision-Language-Action (VLA) models can leverage human data by introducing predictive embeddings as latent actions, moving beyond traditional inverse- or forward-dynamics reconstruction paradigm.
2. The UniHand-Mix dataset and the hybrid pretraining scheme are useful to the community.
3. The paper is well-organized, beginning with clear motivation, followed by progressively detailed explanations of PELA’s architecture.

**Weaknesses:**

1. Absence of real-world validation.
The paper lacks real-world robot experiments to verify that the proposed approach transfers effectively beyond simulation environments.

2. Missing key comparisons and ablations.
Several essential ablations are absent, making it difficult to substantiate the core technical claims:
(1) Without LSP: Train the latent action perceiver end-to-end without the latent state perceiver, while keeping the masked chunk prediction and latent action alignment objectives. This would directly demonstrate the necessity of introducing LSP and the asymmetric EMA update.
(2) Without annotated human videos: Pretrain the model solely with the latent action alignment loss, excluding labeled human data. This would clarify the contribution of annotated videos and illustrate why mapping to ground-truth actions remains important for latent action learning. (This is also a important distinction from LAPA method that do not use hand motion annotations)

3. Lack of formal justification for predictive embeddings.
While the intuition that predictive embeddings capture “motion patterns predictable from context” is appealing, it is neither theoretically formalized nor empirically analyzed. The paper does not provide clear evidence explaining why predictive alignment should generalize better than reconstruction- or masking-based objectives.

4. Unanalyzed latent action space.
The claim that PELA constructs a “unified latent action space” across labeled and unlabeled videos is not supported by analysis. The paper provides no examination of what this space encodes or how well it aligns across domains such as lab, in-the-wild, and robot data.

5. Questionable citation and authorship clarity.
The manuscript references a non-existent work (UniVLA Team. UniVLA: Unified vision-language-action model for cross-embodiment robotic learning. arXiv preprint, 2024a.), raising concerns about citation accuracy and the extent to which large language models may have been involved in the manuscript’s preparation.

[Minor comment]
The reported performance improvements appear modest relative to existing baselines. In particular, according to Figure 3, the model underperforms Being-H0 when trained on the same 1,000+ hours of annotated laboratory data.

**Questions:**

1. The paper claims that combining labeled and unlabeled human videos is crucial. Could the authors provide a variant trained only on unlabeled data using the latent-action alignment loss? How much does annotated data contribute quantitatively to downstream performance (e.g., LIBERO success rate)?
2. Can the authors include or share results for an ablation that removes the Latent State Perceiver (LSP) and trains the Latent Action Perceiver end-to-end with the same losses?
3. The paper states that PELA unifies labeled and unlabeled data in a shared latent space. Could the authors visualize or quantify this alignment?
4. The reference “UniVLA Team, 2024a” appears to correspond to a non-existent paper. Could the authors clarify what work this refers to, or correct the citation?

---

### Official Review · Reviewer_1uvT · 2025-10-31

**Soundness:** 2
**Presentation:** 2
**Contribution:** 2
**Rating:** 4
**Confidence:** 2

**Summary:**

This paper introduces PELA (Predictive Embedding as Latent Action), a framework for scalable pretraining of Vision-Language-Action (VLA) models using both lab-collected and in-the-wild human videos. Instead of relying on heavy reconstruction losses, the authors propose learning predictive embeddings aligned with latent actions derived from inverse dynamics. This design aims to extract transferable motion cues without requiring explicit action labels for all data. The paper also presents UniHand-Mix, a large hybrid dataset (7.5M samples, ~2,000 hours) combining labeled human manipulation data with filtered egocentric videos. The experiments show consistent gains on both hand motion generation and robotic manipulation benchmarks such as LIBERO and RoboCasa.

**Strengths:**

- Clear motivation and solid intuition: The paper tackles a well-known challenge in robot learning (limited high-quality robot data) and proposes a reasonable, well-motivated solution. The idea of “predictive embeddings” feels conceptually elegant and bridges the gap between labeled and unlabeled data in a meaningful way.

- Large-scale hybrid data design: UniHand-Mix is an impressive contribution on its own, and combining precise lab recordings with diverse real-world videos is exactly what this community needs to move toward scalable embodied pretraining.

- Strong empirical results: The model performs very well on both LIBERO and RoboCasa benchmarks, showing that the learned representations transfer effectively. The ablations are thoughtful — for example, the study on wild video ratios and hidden layer selection provides concrete insight into what drives performance.

- Detailed technical presentation: The paper provides a clear formulation of the model, training objectives, and implementation details. It’s also well-structured and visually supported by helpful figures.

**Weaknesses:**

- Questionable novelty at the conceptual level: The proposed framework is essentially a refined combination of existing ideas (latent action modeling, inverse dynamics alignment, and masked prediction). The incremental contribution is in the way these are fused under the “predictive embedding” umbrella.

- Somewhat shallow theoretical grounding: The paper is heavy on empirical validation but light on analysis. It would help to include a clearer intuition (or a small-scale analysis) of why predictive embeddings outperform standard latent actions beyond empirical evidence.

- Data quality concerns: The in-the-wild subset relies on automated annotation and filtering pipelines (e.g., Gemini-2.5-flash, HaWoR). There’s little discussion of the noise level or how label errors impact learning.

- Generalization scope: Experiments are limited to hand-centric tasks. It’s unclear whether PELA generalizes to whole-body motion or broader robotic embodiments.

- Scalability and efficiency: While presented as scalable, the method’s reliance on high-capacity backbones (InternVL3-2B, DINOv3, V-JEPA2) may restrict accessibility for smaller labs. Some efficiency comparison to Being-H0 or LAPA would make the contribution more convincing.

**Questions:**

- How sensitive is the method to the choice of alignment layer (e.g., 19th)? Could this be learned or dynamically selected?

- Does the asymmetric EMA update between LAP and LSP significantly impact stability? What happens if it’s removed?

- Can predictive embeddings extend beyond hand motion — say, to full-body human videos or other action domains?

- How do annotation errors in the in-the-wild subset affect training quality?

- What’s the compute footprint compared to other 2B-scale VLA models like Being-H0 or UniVLA?

---

### Official Review · Reviewer_1Hgc · 2025-11-03

**Soundness:** 4
**Presentation:** 3
**Contribution:** 4
**Rating:** 8
**Confidence:** 3

**Summary:**

This paper proposes PELA, a pretraining framework for scalable Vision-Language-Action (VLA) models using both lab-annotated and in-the-wild human videos. The central idea is to move beyond full reconstruction of dynamics and instead align predictive embeddings (hidden states of a pretrained VLM) with latent actions derived from short-horizon visual transitions. The authors introduce two paired modules: Latent Action Perceiver (LAP) and Latent State Perceiver (LSP), that extract transition-level dynamics from boundary frames and context features, coupled via asymmetric EMA updates for stability. A new 7.5M-sample dataset, UniHand-Mix, combines labeled and unlabeled human manipulation videos for hybrid pretraining. The pretrained model is transferred to robot control tasks via a flow-matching head, achieving state-of-the-art results on LIBERO and RoboCasa benchmarks among sub-3B-parameter models.

**Strengths:**

The paper offers a novel latent-action formulation that connects inverse-dynamics-like latent supervision with predictive hidden states inside a VLM. The dual-Perceiver (LSP/LAP) design and EMA update mechanism are original contributions addressing instability when aligning heterogeneous modules. The hybrid data strategy and pretrain to flow-matching transfer pipeline are well-executed. The authors present ablations showing clear trends (e.g., scaling with in-the-wild data, layer selection effects), and comparisons against strong recent VLAs such as GR00T N1.5 and UniVLA demonstrate interesting gains.

**Weaknesses:**

Clarification on Training:
The paper does not explicitly spell out which parameters are optimized concurrently during pretraining, e.g. whether predictive embeddings (VLM backbone) and LAP/LSP modules are co-trained end-to-end or updated under alternating schedules. The EMA coupling is described conceptually but lacks an algorithmic summary or pseudocode.

Ablation interpretation:
The w/o-alignment variant performs surprisingly well (85.8 % on LIBERO), close to the full model, suggesting either that MCP already encodes substantial motion priors or that alignment gains are task-dependent. Clarifying what fraction of unlabeled data contributes to the gap would help.

**Questions:**

1. Are LAP and LSP co-optimized in the same update step with shared gradients, or alternately updated with EMA synchronization? Are VLM backbone parameters (from InternVL3-2B) fully trainable or partially frozen?

2. The “w/o alignment” variant still performs strongly. Can the authors comment on what specific benefits alignment brings (e.g., improved robustness to in-the-wild data, faster convergence) beyond a small quantitative gain?

3. Why do you believe latent actions derived from human hands transfer effectively to robot arms and grippers? Some supporting analysis or visualization would strengthen this claim.

---

### Note · Authors · 2025-11-14

I have read and agree with the venue's withdrawal policy on behalf of myself and my co-authors.